# A Review of the Release Profiles and Efficacies of Chemotherapy Drug-Loaded Electrospun Membranes

**DOI:** 10.3390/polym15020251

**Published:** 2023-01-04

**Authors:** Zhenyu Lin, Hao Chen, Jiawei Xu, Jie Wang, Huijing Wang, Shifen Huang, Shanshan Xu

**Affiliations:** 1Institute for Advanced Study, Shenzhen University, Shenzhen 518060, China; 2Key Laboratory of Optoelectronic Devices and Systems of Ministry of Education and Guangdong Province, College of Physics and Optoelectronic Engineering, Shenzhen University, Shenzhen 518060, China

**Keywords:** local chemotherapy, electrospun membranes, controlled release, release profiles, biodegradable polymers

## Abstract

Electrospun fibrous membranes loaded with chemotherapy drugs have been broadly studied, many of which have had promising data demonstrating therapeutic effects on cancer cell inhibition, tumor size reduction, the life extension of tumor-bearing animals, and more. Nevertheless, their drug release profiles are difficult to predict since their degradation pattern varies with crystalline polymers. In addition, there is room for improving their release performances, optimizing the release patterns, and achieving better therapeutic outcomes. In this review, the key factors affecting electrospun membrane drug release profiles have been systematically reviewed. Case studies of the release profiles of typical chemotherapy drugs are carried out to determine the preferred polymer choices and techniques to achieve the expected prolonged or enhanced release profiles. The therapeutic effects of these electrospun, chemo-drug-loaded membranes are also discussed. This review aims to assist in the design of future drug-loaded electrospun materials to achieve preferred release profiles with enhanced therapeutic efficacies.

## 1. Introduction

The International Agency for Research on Cancer (IARC) has estimated over 19 million new cancer cases and almost 10 million deaths from cancer based on the 2020 global cancer statistics (GLOBOCAN) database, which covers most of the countries in the world [1]. Cancers are well known for their ability to metastasize, which leads to irreversible and fatal progressions. The general strategy against solid carcinoma is surgical removal with subsequent chemotherapies or radiotherapies to clean up cancer residues. However, there could be hardly any chances of operation for some carcinomas located in sensitive areas like the spinal cord or brain. In addition, chemotherapies are toxic for the whole body and cannot concentrate at specific sites; radiotherapies are effective at specific areas, however, without the ability of continuous application for a prolonged period.

These situations brought out the demands for better local therapeutic measures to ultimately limit the reoccurrence rate of cancers and inhibit tumor growth in sensitive areas where operations were seldom applicable. A series of local cancer therapies, like tumor sclerosis, tumor embolization, and intra-tumoral injection, were developed [2]. In addition to these routine procedures, drug-loaded hydrogels, electrospun membranes, and other in situ delivery systems have great potential against local carcinoma since they are excellent in local tissue adhesion with high porosity that facilitates drug delivery and cell attachments [3,4]. Hydrogels usually degrade in a relatively short period due to their high water content, making them applicable for localized short-term release (usually less than a month) of drugs [5,6,7]. Electrospun membranes are dry materials with more reliable and tunable mechanical properties that are compatible with local tissue. They possess the ability of both short-term and prolonged release, making them more suitable for a variety of situations in local chemotherapy.

Electrospun membranes could be fabricated using various techniques with numerous combinations of parameters. These parameters include polymer choices, solution preparation methods, fabrication techniques, drug types/loading methods, etc. All of them could affect the drug release profiles of electrospun membranes. Nevertheless, these parameters have not yet been systematically determined, confusing many researchers.

This review provides insight into the key factors affecting the release profiles of the implantable electrospun membranes tailored for local carcinoma. Their release patterns and therapeutic effects are categorized and concluded to assist in the future design of drug-loaded electrospun delivery systems.

## 2. Release Profiles of the Chemotherapy Drug-Loaded Electrospun Membranes

### 2.1. Key Factors Affecting the Release Profiles

#### 2.1.1. Polymers

Synthetic biodegradable polymers were applied broadly in clinical implantable products like bone/ligament fixing screws [8,9,10,11], pins [12,13], sutures [14,15,16,17], scaffolds [18,19], degradable embedded thread [20,21,22] in plastic surgeries, and more, as illustrated in Figure 1. They possess highly controllable parameters like monomer types, combinations and arrangements, chain length, and a molecular weight readily tailored for various clinical needs. These polymeric products’ degradation durations could range from weeks to months [23] or even a few years [24]. It is essential to determine the degradation characteristics of these polymers in their crystalline forms since they are closely related to their corresponding drug release patterns. Furthermore, because of the high surface-volume ratio and loss of crystallinity, biodegradable polymers in electrospun materials tend to degrade more rapidly with changes in their mass loss pattern [25,26,27]. As a result, it is necessary to investigate the representative research examples on the degradation of electrospun implantable polymers to retrieve the real situations of their degradation patterns rather than simply adopting the situations of crystalline biodegradable polymers.

PLA is the most thoroughly studied and broadly applied biodegradable polyester [32]. It represents a group of polymers, including PLLA (L-lactic acid as the only monomer), PDLA (D-lactic acid only), and PDLLA (a copolymer of L- and D-lactic acid), because of the chiral structure of lactic acid. The degradation pattern of PLA appeared as a long-term tri-step process: (1) hydrolysis of ester “joints” accompanied by significant loss of molecular weight (Mw) and increasing swelling; and (2) progressing water interaction with crystalline “long bones” and reaching the peak stage of swelling; (3) Further degradation into oligomers and molecules of lactic acid caused the system to gradually disintegrate into the aqueous environment [33,34]. Degradation of non-electrospun PLA in human physiological conditions could take up to a year or more [35,36,37]. In general physiological conditions (PBS 1X pH 7.4 at 37 °C), it was reported that both low and high molecular weight (Mw) PLA took around 10 weeks to reach a mass loss of over 5% [33]. However, the weight loss of electrospun PLA membranes was reported to be around 5–12.5% in just 4–6 weeks [25,26,27]. Apparent weight loss, decrease in electrospun fiber diameters, and shrinkage in volume would likely occur in about three months, which was faster than the standard implantable products made with relatively crystalline and bulky forms of PLA [38,39].

Polyglycolic acid (PGA) is another FDA-approved biodegradable polymer utilized widely in the medical field. It is the most hydrophilic among the standard medical polyesters [23]. Glycolic acid units lack a methyl group compared to lactic acid, resulting in significantly higher hydrophilicity [35] and thus degrading relatively faster than PLA. Unlike PLA, the degradation pattern of PGA appeared to be a two-step process overall. Although polyesters generally degrade in three stages: breaking down of amorphous ester linkages “joints” as the first step (unobvious mass loss), then gradual hydrolysis of the crystalline tightly packed regions “long bones” (slight mass loss) followed by a prolonged process of complete degradation to oligomers and monomers (obvious mass loss). However, due to the hydrophilic nature of glycolic acid units, the hydrolysis of “joints” and “long bones” occurred nearly at the same pace, so that the first two stages blend together [40,41]. PGA has a glass transition temperature (T_g_) of about 36 °C, just close to human body temperature, which allows it to be applied in suture material or other tissue engineering equipment that needs faster degradation. Degradation of medical PGA implants like scaffolds and stents is relatively fast, taking just a few months [42]. Observations on the PGA sutures revealed that 70% degradation occurred in about 3 months, with half the crystallinity lost [41]. Compared to crystalline forms, electrospun PGA tends to degrade even faster. Studies demonstrated that over half of the mass loss and volume reduction had occurred in electrospun PGA in less than a month in an aqueous physiological environment [43,44].

Polylactide-co-glycolide (PLGA) is a polymer that combines the many advantages of both PLA and PGA. As a copolymer, PLGA allowed flexible tuning of the ratio and arrangement of lactide and glycolide units, resulting in a more tunable degradation time, mechanical properties, and hydrophilicity. Medical implants made of PLGA, like bone scaffolds [45], cartilage repair scaffolds [46], cell seeding foams [47], nerve recovery guide material [48], and many more, were able to last up to one year or even longer [49,50,51]. In comparison, the degradation duration of electrospun PLGA 50/50 membranes appeared to be between PLA and PGA. The overall mass loss tipping point appeared after around one month. The mass drop turning point for PLGA with a higher LA ratio of 75:25 appeared after approximately two months. During this period, the fibers kept swelling and gradually became more intercrossing and “melted”, finally into clusters of agglomerates [52]. PLGA tends to have a tri-stage degradation process. At the first stage, both “joint” hydrolysis and loosening of the crystalline area (due to hydrophilic glycolide units attracting more water) resulted in a shorter first stage than PLA but not as dramatic as PGA. The second phase of PLGA degradation was dramatic since the “long bone” area of PLGA was amorphous, which helped water molecules invade and hydrolyze. The third phase is a prolonged process of destruction of oligomers and monomers similar to that of other polyesters [44,53].

Polycaprolactone (PCL) is another extensively utilized biodegradable polyester. It is semi-crystalline [54,55] with an apparently longer degradative duration than the three polyesters mentioned above; therefore, it was applied less than them. However, the long-term degradation property of PCL is desirable for local chemotherapies, which require prolonged delivery of chemotherapy drugs to inhibit and eradicate carcinoma cells. Complete degradation of PCL capsules in vivo could take up to two years [55]. The electrospinning technique interrupts the crystallinity of PCL, thus resulting in a faster degradation rate. The detailed degradation profile could also be affected by electrospun parameters. It was reported that less than 3% weight loss of electrospun PCL (80 kD) had occurred after 42 days [25] or about 15% loss at 30 days of electrospun PCL (70 kD). In general, physiological conditions, about a 30% decline in fiber diameter occurred after 107 days [56].

Chitosan is a polymer generated by the deacetylation of an abundantly seen natural chitin. It has a number of advantages, including affinity for organ tissue surfaces [57], well-recognized swelling ability [56] and antibacterial activities [57], which are excellent properties for the design of drug releasing biodegradable implants. The range of Mw of currently produced chitosan could be from 3800 to 2 million Da with deacetylation rate from 66 to 95%, and the degradation rate is usually proportional to its degree of acetylation [58]. It is also sensitive to biological (bacterial), chemical (hydrolytic), biochemical (lysozymes), and physical (temperature) degradations. A study reported that, under general physiological conditions (sterile), hydrolytic degradation had a noticeable impact on chitosan’s molecular weight within 3 days of the experiment. At 28 days after the end of the experiment, the Mw of chitosan had dropped by 65.6%, while the Mw declined by 81.9% compared to the state right after sterilization [58]. It could be inferred that the hydrolytic degradation process of chitosan in electrospun fibers would be even faster due to the enhanced surface contact area with water molecules. The hydrophilic nature of chitosan makes it an excellent candidate to be blended with hydrophobic synthetic polymers to reduce their overall degradation duration while helping to avoid burst release of hydrophilic drugs and to tune their drug release profiles towards logistic or even linear patterns [59,60]. A summary of degradation features and intended usage of PLA, PGA, PLGA, PCL and chitosan was demonstrated in Table 1.

#### 2.1.2. Techniques of Electrospinning

Blend electrospinning is the most straightforward fabrication technique for drug-loaded fibrous membranes (Figure 2a). All the membrane components are blended in proper solvent systems and electrospun in one step. Additionally, it has the least requirement for fabrication set-up by using regular needle tips, making it the most cost-effective technique for scale-up processing. Since all the membrane components are dissolved in one step, compatibility among the drugs, polymers, and solvent systems is the key to desirable controlled release profiles. If they are not compatible and usually appear as a turbid or even phase-separated polymer blend, the drugs are highly likely to accumulate on the surfaces of electrospun fibers, leading to the burst release of drugs [73,74,75,76]. A properly arranged blending of electrospun components would yield homogeneous drug-load membranes that provide desirable controlled release profiles. Under this circumstance, their drug release profiles usually followed the degradation pattern of their polymer matrix. However, drug molecules located deep inside the polymer fibers could be tightly trapped and maintain a stationary phase of drug release for a prolonged period of time (second stage) until the polymer structure began to disintegrate [71,77]. A study reported the release profile and mechanisms of electrospun PLGA loaded with ciprofloxacin hydrochloride. The release profile was demonstrated to be a logistic tri-stage pattern for over 40 days [53], in accordance with the PLGA degradation process explained above. Another study incorporated doxorubicin (DOX) within PLA/PEG blended membranes; their general release profile of DOX followed the characteristic two-stage logistic degradation pattern of PLA. In contrast, the incorporation of 20% PEG within their PLA membranes adjusted the polymer system polarity significantly and thus became relatively compatible with DOX and achieved an excellent 30 day prolonged release of ~85% of loaded DOX [61].

Coaxial electrospinning (Figure 2b) is a technique for fabricating core-shell fibers whose structure is featured in the inner and outer layers like electric wires. This type of fiber can achieve special functions that regular blending of electrospun fibers could hardly accomplish. Typical applications of the core-shell fibrous membrane include protecting active chemicals vulnerable to light or oxidative impacts, providing a special control-release effect for hydrophilic drugs, etc. This technique utilized coaxial needle tips connected to two supplies of polymer solutions. The electric field would be applied to both solutions simultaneously, yielding electrospun fibers with core and shell structures. Coaxial electrospun fibers allow for the delicate utilization of the incompatibility between hydrophobic polymers and hydrophilic drugs. They entrap the hydrophilic drugs inside the core to avoid direct contact with water, which causes burst release. The shell structure forced the hydrophilic drugs to release, following the degradation patterns of the shell polymers. Studies utilized a core/shell combination of chitosan/PLA to achieve prolonged release of DOX to 30 days [59] and PVA/PCL to extend the release of 5-fluorouracil (5-FU) to 25 days [78]. A core-shell structure also overcame the tight entrapment of the hydrophilic drug in the blending technique mentioned above and boosted the final drug release to a high ratio of 80% [59,78].

Another path to achieving core-shell structure is to prepare a polymer blend as an emulsion, called “emulsion electrospinning” (Figure 2c). This measure possesses the same advantage of one-step formation as blend electrospinning. At the same time, it avoids using coaxial needles that are difficult to produce and easily get stuck by polymer solutions during electrospinning processing, which are fatal issues for upscaling productions. Emulsion electrospinning is attained by preparing an emulsion of polymer, drug, and surfactant. The hydrophobic polymers were dissolved in non-polar organic solvents to form the oil phase, while the hydrophilic drugs and the surfactants were mixed to form micelles in the aqueous phase. When the stable emulsion system is being electrospun, the oil-phase solvent evaporates quickly, increasing the viscosity and surface energy and separating the two phases, which creates a core-shell structure [77,79]. From the release profile point of view, emulsion electrospinning combines the advantages of blending electrospinning (straightforward and cost-effective for upscaling) and coaxial electrospinning (overcomes the difficulties of incorporating hydrophilic drugs in hydrophobic polymers). It seems the most desirable for prolonged controlled release of hydrophilic chemotheraphy drugs. Nevertheless, due to the incorporation of surfactants, allergy issues should be highly concerning during in vivo applications.

Surface modification is an extension of electrospinning techniques with multiple advantages (Figure 2d). It overcomes the incompatibility issue between polymer solvent systems and the drugs by separating the drug-loading and electrospun membrane fabrication processes while preventing unwanted reactions or degradation of drugs in some solvent conditions. Surface modification to electrospun membranes was usually carried out in two ways: absorption and cross-linking. The absorption method utilizes the porous/high surface area/swelling nature of many electrospun fibers to absorb drugs with the help of an electrostatic force and concentration gradient. Nevertheless, the absorption type of drug membranes often demonstrated a small burst release at the beginning, followed by a prolonged release, since the electrostatic force was not strong enough to create a tight bond between drugs and polymer fibers [80]. Cross-linking was usually a better option for more controllable release profiles with suppressed burst release, a higher drug release ratio, and a longer duration [81]. However, the safety of cross-linker chemicals remains a concern when applied to the surface of human organ tissue. In addition, surface modification processes are often sensitive to pH, ionic strength, temperature fluctuations, etc. The drug loading process often took longer than other techniques, and the loading amount was not entirely controllable and frequently required extra procedures to evaluate actual drug content.

### 2.2. Investigating the Release Profiles of Drug-Loaded Electrospun Membranes

Although the key factors affecting the drug release from electrospun membranes are clear, situations become complicated when multiple variables are combined. In order to make a practical analysis from the sea of related research, we selected the release data targeting four types of representative cancer drugs (5-FU and DOX as the representative hydrophilic chemotherapy drugs, and PTX and CIS as the hydrophobic ones), and refined those results to match two major aims: prolonged control-release for hydrophilic chemotherapy drugs and enhanced release for hydrophobic chemotherapy drugs. The release profiles of electrospun membranes loaded with typical hydrophilic/hydrophobic chemotherapy drugs that successfully achieved these aims are collected and sorted in Table 2 with discussions, respectively.

#### 2.2.1. Challenges in the Drug Release of Chemotherapy Drug-Loaded Electrospun Membranes

Researchers have fabricated delicate electrospun materials with anti-tumor or even multiple functions. Unfortunately, some did not achieve satisfying release profiles that might decrease their therapeutic efficacies and shorten their anti-cancer time range. The issue mainly focused on incompatible drug-polymer systems. Implantable hydrophobic polymers like PLA, PLGA, and PCL possess excellent mechanical properties and stability that are attractive to many researchers. However, when blended with hydrophilic drugs, the polar incompatibility will cause an uneven distribution of drugs in the polymers. A part of the drug would accumulate on the surface of the electrospun materials (Figure 3, DOX-loaded membrane) and cause burst release; the remaining drug would frequently be trapped tight inside the polymers, which would be unable to release for a very long time before significant degradation of the polymers [73,74,90,91] (as demonstrated in Figure 4). Burst chemotherapy drug release could cause acute toxicity in the test animals and reduce the effective period. It is worthy of optimization. Another issue was sometimes seen in hydrophobic drug-loaded membranes: the overall drug release amount in the aqueous system tends to be low [92]. This issue would lead to a wasteful use of a portion of chemotherapy drugs, which would increase the cost of manufacturing and raise the product’s price.

#### 2.2.2. Strategies to Enhance the Release Profiles of Hydrophilic Chemotherapy Drug-Loaded Electrospun Membranes

Hydrophilic chemotherapy drugs share properties like fast effects through rapid concentration elevation and short retention times in the human body. In order to maintain a specific local drug concentration, these chemotherapy drugs had to be applied repeatedly in a short period, which increased the suffering of patients; furthermore, a rapid elevation of drug concentration is closely related to acute toxicity events. As a result, electrospun membranes for local chemotherapies should be tuned to possess prolonged logistic and even linear release profiles for hydrophilic chemotherapy drugs, reducing their burst release to achieve reliable drug effectiveness.

It is critical to match the polarity between drugs and polymers to achieve proper controlled release of hydrophilic chemotherapy drugs from the electrospun membrane. Nevertheless, most long-term degradative polymers are relatively hydrophobic and thus natively incompatible with hydrophilic chemotherapy drugs. This contradiction has presented challenges for researchers. To address this problem, polymer systems for electrospinning membranes should be tuned to be relatively less hydrophobic and easier to degrade in water. The strategies include incorporating amphiphilic chemicals like low- to mid-Mw PEG that is mostly safe for human tissue [61] and adding polar polymers into the system to increase overall polarity like PGA, PVA, PVP, chitosan, gelatin, etc. [83,84,85]. PLA supplemented with hydrophilic polymers like gelatin [80,84], PEG [61], and PLGA 50:50 with PVA [83] demonstrated successful prolonged release profiles for hydrophilic chemotherapy drugs DOX and 5-FU, allowing them to switch from the profile of burst release in 24 h to a prolonged logistic release for more than a week or even 10 days. However, this short-term release via blending electrospinning techniques could not always provide enough therapeutic duration. In order to achieve a longer release time of around a month, core-shell electrospinning techniques would be more desirable. By designing a hydrophilic core solution of hydrophilic drugs with chitosan or PVA together with a shell solution of hydrophobic long-term biodegradable polymers like PLGA 50:50, PLA, or PCL, the release of DOX or 5-FU could be prolonged to 25~30 days [59,78]. Conjugation of hydrophilic drugs prior to electrospinning could even achieve a linear release that is valuable for clinical practices [59].

#### 2.2.3. Strategies to Enhance the Release Profiles of Hydrophobic Chemotherapy Drug-Loaded Electrospun Membranes

The trickiest thing about applying hydrophobic chemotherapy drugs is increasing their solubility and bioavailability. Raw hydrophobic drugs are usually crystalline particles that precipitate in an aqueous environment without entering the cancer cells. Electrospun membranes should be designed to largely enhance their release quantity over the polymer degradation period and act as carriers to increase their compatibility with an aqueous environment. PLA, PLGA 50:50, and PCL electrospun membranes could achieve logistic release of more than 50% encapsulated PTX or cisplatin for 30 days or longer [60,63,67,68,71,87,89]. Complex encapsulation techniques like conjugation [59], dextran encapsulation [87], blending with chitosan [60], and release enhancers [71] were effective in boosting release efficiency (released drug ratio). In addition, some of these carriers could help achieve linear release of PTX or cisplatin, which is significantly important for providing reliable therapeutic outcomes that require a stable drug administration dose [60,71,87].

## 3. Therapeutic Effects

Properly tuned electrospun chemotherapy drug membranes with prolonged and controlled release profiles were proven effective against the activities of cancer cells in vitro and inhibited the growth of local tumors while extending the lives of tumor-bearing animals. For in vitro cell toxicity studies, DOX-loaded PCL/PLA-based controlled release membranes were mostly reported to exhibit antiproliferative effects against the MCF-7 human breast cancer cell line [59,70,82,103] and were found to be modulating BCL-2 and caspase-3 signaling pathways of MCF-7 in synergism with berberine [70]. More prolonged-releasing DOX membranes were also reported effective against human glioblastoma U-87 MG [80], lung cancer A549 [61], and fibroblast 3T3 [104] cell lines, etc. The use of 5-FU controlled release membranes was found to be effective against human colorectal carcinoma HCT-116 cells [62], late-stage human breast cancer MDA-MB-231 cells [78], and more. PTX-loaded enhanced releasing membranes were mostly studied against breast cancer cell lines and found to be effective [59,67]. Cisplatin-loaded membranes demonstrated antiproliferative effects against mouse lung cancer LLC cells [71], human lung cancer spc-a-1 cells [64], C6 glioma cells [89], murine ovarian surface epithelial cells ID8 [105] and mouse fibroblasts NCBI C161 [88]. A study of delicately fabricated triaxial electrospun membranes loaded with DOX, 5-FU, and PTX achieved a significantly better antiproliferative effect against MCF-7 human breast cancer cells than the routine single-layer membrane [59].

For in vivo anti-tumor studies, a local application of DOX-loaded controlled release membranes was reported to reduce glioblastoma by nearly 80% at day 20, borne by naked mice [80]. Another study reported that a DOX-micelle-loaded electrospun membrane could achieve approximately 18% accumulation in tumor tissue in three days in MCF-7 breast tumor-bearing mice, which was more than a six-fold higher local DOX accumulation compared to intravenously injected DOX and demonstrated low systemic toxicity [103]. A PCL/PLGA bilayer membrane loaded with PTX and locally applied to human esophageal carcinoma cells (Eca109) injected into naked mice was reported to maintain a superior 35 day survival rate over the intravenous injection groups and control groups. Their PTX membrane group also demonstrated lower systemic toxicity than PTX intravenous injection, as reflected by a higher body weight [86]. A study on the anti-glioblastoma effect of PTX-dextran-PLA scaffolds on naked mice found improvements in overall survival on scaffolds with varied release profiles [87]. A cisplatin-loaded PCL/chitosan membrane was reported to reduce cervical tumor volume significantly more in Swiss albino mice over 14 and 21 days compared to applying plain cisplatin [60]. Cisplatin-loaded PLGA 50:50 electrospun membranes were also found to deliver significantly more drug in the locally treated area (brain) than systemic plasma for over eight weeks with no apparent inflammatory responses [68].

Unfortunately, there are not yet any FDA-approved products of electrospun chemo-drug-loaded membranes on the U.S. market. There are currently two electrospun material products listed on the clinical trial database of the U.S. National Institute of Health (NIH). One is an antibacterial electrospun material loaded with ciprofloxacin (CIP), metronidazole (MET), and minocycline (MINO) for immature necrotic teeth (ClinicalTrials.gov Identifier: NCT03690960). The other is a tissue repair scaffold for treating dermatologic wounds created by the surgical removal of non-melanoma skin cancers (ClinicalTrials.gov Identifier: NCT02409628). Hopefully, there will be chemotherapy drug-loaded electrospun material products available for clinical trials against local carcinoma.

## 4. Conclusions

In this review, the key factors affecting the release profiles of chemotherapy drug-loaded electrospun membranes were systematically discussed with mechanistic elaborations and case studies. Research on electrospun membranes loaded with several representative chemotherapy drugs (DOX, 5-FU, PTX, and cisplatin) with varied polymers and techniques are selected and categorized to demonstrate the reality of drug release and to provide meaningful experiences towards desirable release profiles according to varied needs. Their therapeutic evidence, with both in vitro antiproliferative cell studies and in vivo local anti-tumor effects, was also refined and demonstrated. Hopefully, this review could serve as a strategic guide for the design of drug-loaded electrospun membranes, especially to be applied as implants against local carcinoma, to unravel the relationship between elements of drug membranes and their overall release profiles.

## Figures and Tables

**Figure 1 polymers-15-00251-f001:**
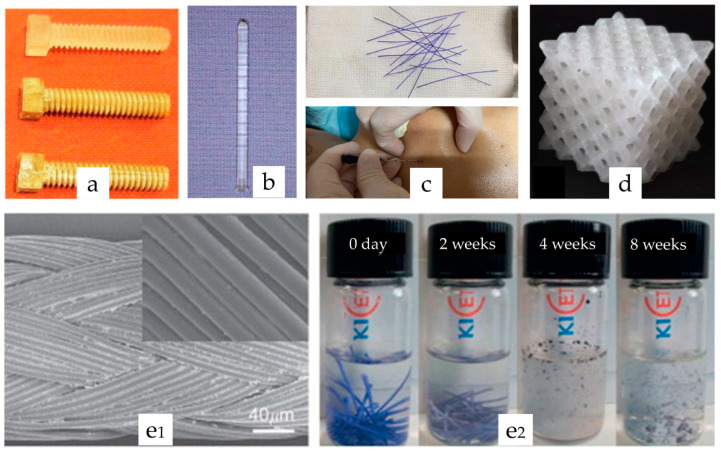
Biodegradable polymeric implants. (**a**) Bone fixation screws made of polylactic acid (PLA) with phosphate enforcement (reprinted from [9] with copyright permission). (**b**) Biodegradable 2 mm polylactide bone pins (reprinted from [28] with copyright permission). (**c**) Polylactide-co-glycolide (PLGA) embedding thread (reprinted from [29] under the Creative Commons CC-BY-NC-ND license). (**d**) PLA scaffold (reprinted from [30] under the Creative Commons CC-BY license). (**e1**) PLGA suture SEM image, and (**e2**) degradation process (reprinted from [31] with copyright permission).

**Figure 2 polymers-15-00251-f002:**
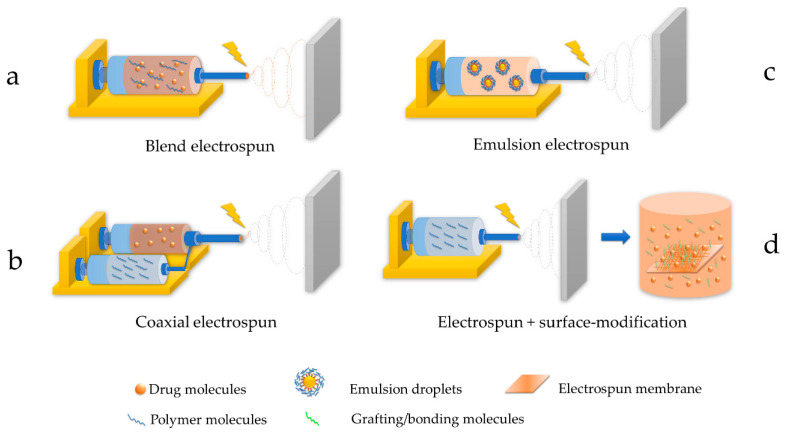
Typical electrospinning techniques to fabricate drug-loaded membranes including (**a**) blend electrospun (direct electrospun of polymer-drug blend). (**b**) coaxial electrospun (drug-core formulation and polymer-shell formulation being electrospun independently). (**c**) emulsion electrospun (electrospun of a thermodynamically stable drug-polymer formulation) (**d**) electrospun + surface modification (drug was absorbed or grafted to an electrospun film as a secondary process).

**Figure 3 polymers-15-00251-f003:**
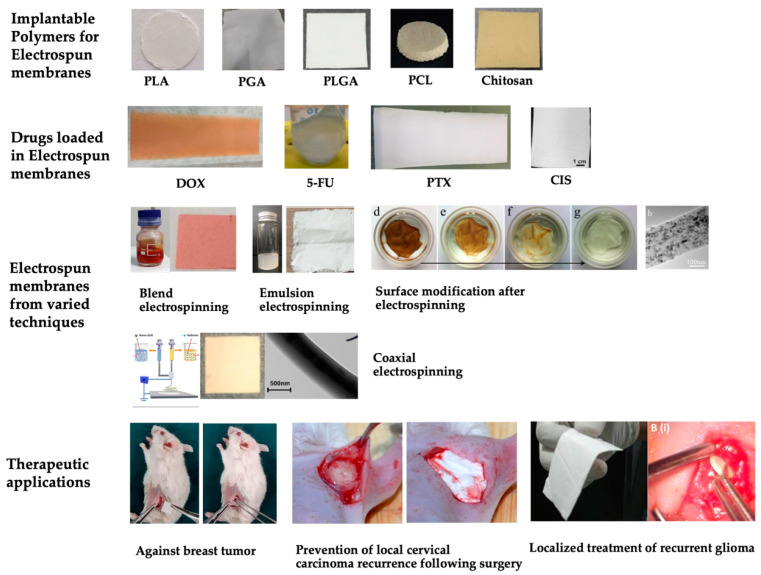
Examples of electrospun membranes fabricated with varied implantable polymeric materials (PLA [93], PGA, PLGA [94], PCL [95], chitosan [94]), loaded drugs (DOX, 5-FU [96], paclitaxel (PTX), and cisplatin (CIS) [71]), techniques (blend electrospinning, emulsion electrospinning [97], coaxial electrospinning [98], and surface modification [99]), and therapeutic applications (electrospun piperine loaded PCL against breast tumor [100], electrospun oxaliplatin loaded PLLA preventing local cervical carcinoma recurrence [101], and electrospun temozolomide loaded PLGA-PLA-PCL for localized treatment of recurrent glioma [102]). Images are reprinted with copyright permissions or under the Creative Commons CC-BY license.

**Figure 4 polymers-15-00251-f004:**
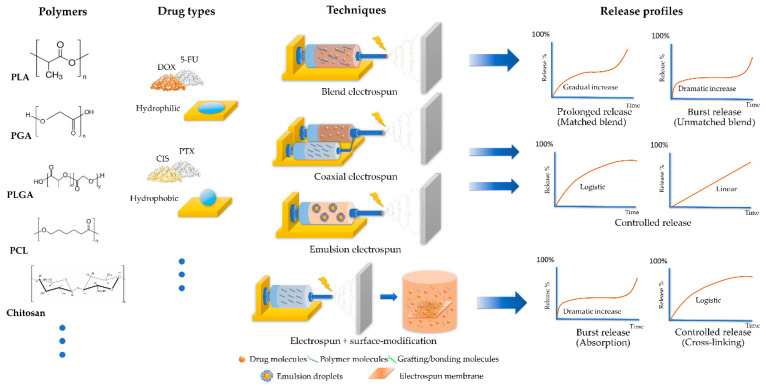
Key factors (polymers, drug hydrophilicity, and electrospinning techniques) affecting the release profiles of chemotherapy drug-loaded electrospun membranes. Combining or using surface-absorbing drugs with hydrophilicity-mismatched polymers would result in a burst release followed by a long stationary phase. Proper matching of drug-polymer hydrophilicity or adapting proper techniques would result in a gradual release with profiles close to logistic or linear increase.

**Table 1 polymers-15-00251-t001:** Summary of typical implantable polymers for electrospinning.

Polymer	Degradation Duration in Crystalline Forms	Degradation Duration in Electrospun Forms	Hydrophobicity	Intended Use
PLA 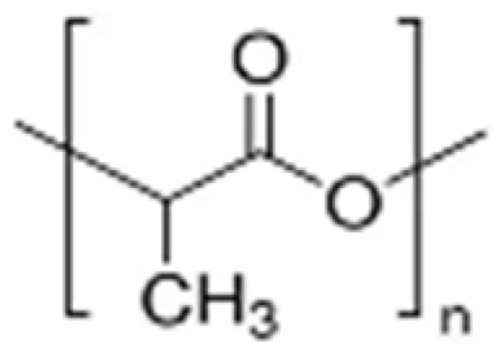	More than one year [35,36,37].	Approximately three months, less than half a year [38,39].	Hydrophobic.	Long-term structural materials [9,28] and delivery systems [61,62,63,64].
PGA 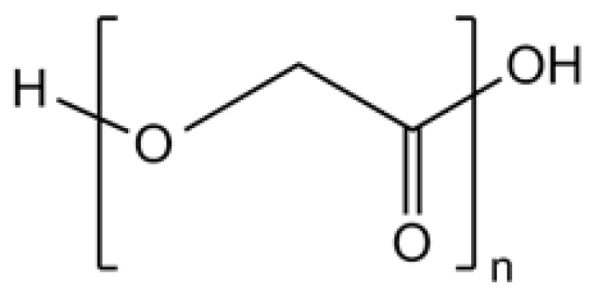	A few months [41,42].	Approximately or less than one month [43,44].	Hydrophilic.	Tissue engineering material [35,43]. and tuning down the degradation period of long-term delivery systems [65,66].
PLGA 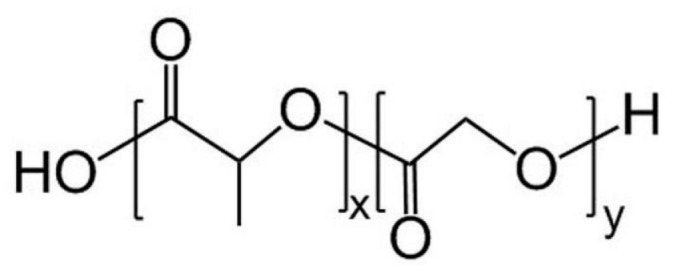	More than one year [49,50,51].	Approximately two months [52].	Hydrophilicity proportional to the GA ratio.	Mid-term flexible structural materials [29,31] and delivery system [67,68].
PCL 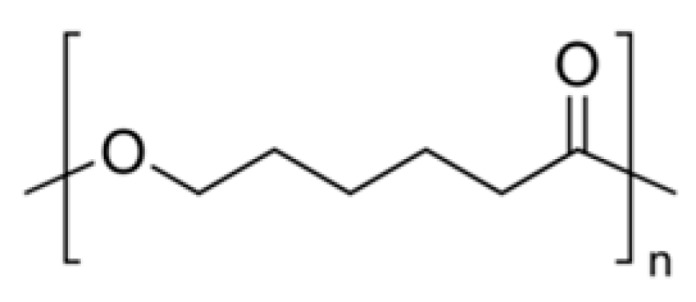	Up to two years [55].	A 30% decline in fiber diameter after around three months [69].	Hydrophobic.	Long-term structural materials [25,69] and delivery system [70,71].
Chitosan 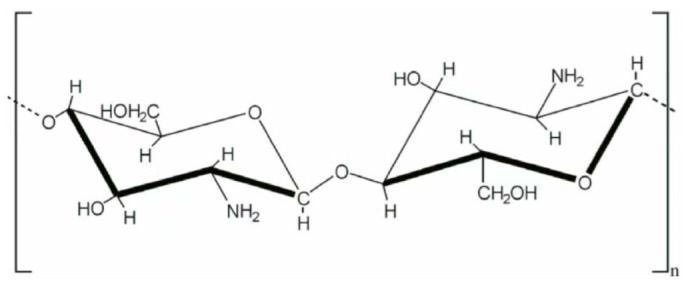	Approximately one month [58].	Limited amount of whole process degradation data.	Hydrophilic.	Short-term drug delivery system and tuning down the degradation period of other long-term delivery systems [59,60], specialty in antibacterial ability [57], and pH-responsive delivery [72].

**Table 2 polymers-15-00251-t002:** Polymer information, electrospun techniques, and release profiles of electrospun chemotherapy drug membranes with controlled release properties.

Chemo Drug	Polymer Information	Techniques	Release Profile and Maximum Released %
DOX	PCL 80 kD	Blending	~9 days logistic to ~30% [70]
	Shell PCL 80 kD + Core PEO 300 kD	Coaxial	~10 days logistic to ~60% [82]
	PLA 380 kD + Gelatin (Type A from porcine skin, 210–250 g Bloom)	Immersion Coating	~6 days logistic to ~30% [80]
	PLA 186 kD + PEG 4 kD	Blending (High Shear) with an enhancer	30 days exponential to ~85% [61]
	PLGA 50:50 81 kD + branched PEI 25 kD + PVA 85–124 kD	Surface Mod Coating	50 days logistic to ~50% [83]
	Core Chitosan (75–85% deacetylated 200 kD) + Shell PLA 150 kD	Coaxial Tri-Layer + Complex Encapsulation	~30 days logistic to ~80% [59]
5-FU	PLA pellets 200 kD	Complex Encapsulation	~4 days logistic to ~90% [62]
	PLGA 50:50 10 kD + gelatin (type A from porcine skin, 300 bloom)	Blending	~10 days logistic to ~90% [84]
	Shell PCL 80 kD + Core PVA 89–98 kD	Coaxial	~25 days logistic to ~80% [78]
	Core PVP + Shell PCL 70% + PVP 30% PCL 80 kD, PVP 360 kD	Coaxial	~7 days logistic to ~80% [85]
	Core Chitosan (75–85% deacetylated 200 kD) + Shell PLA 150 kD	Coaxial Tri-Layer	~30 days linear to ~80% [59]
PTX	PCL 200 kD + PLGA 75:25 80 kD	Bi-layer	~9 days logistic to ~35% [86]
	Core Chitosan (75–85% deacetylated 200 kD) + Shell PLA 150 kD	Coaxial Tri-Layer + Complex Encapsulation	~21 days logistic to ~80% [59]
	Dextran 450–650 kD + PLA 260 kD	Blending	~35 days linear to ~90% [87]
	PLA + Chitosan	Blending	14 days logistic to ~30% [88]
	PLA	Blending	~40 days logistic to ~80% [63]
	PLGA 50:50 33 kD	Blending	~42 days logistic to ~80% [67]
CIS	PCL 45 kD + Chitosan 310 kD	Blending	30 days linear to ~65% [60]
	PCL 70–90 kD	Blending + Release Enhancer	~70 days linear to ~60% [71]
	PLA 85–160 kD + PLGA 50:50 50–70 kD	Blending	~33 days logistic to ~70% [89]
	PLGA 50:50 33 kD	Blending	~30 days exponential to ~100% [68]
	PLA 100 kD	Blending	~11 days logistic to ~45% [64]

## Data Availability

Not applicable.

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
