# Peer review of "A Review of the Release Profiles and Efficacies of Chemotherapy Drug-Loaded Electrospun Membranes"

_polymers, 2023, doi:10.3390/polym15020251_

Round 1
Reviewer 1 Report
In this review, the authors reported chemotherapy drug-loaded electrospun membranes' release profiles and efficacies. The author first showed the key factors affecting the release profiles, and then reported the advantages and disadvantages of four classic methods and technologies. Then, four representative cancer drugs were used to study the release characteristics of the supported electrospun membrane. Finally, he verified the therapeutic effects with both in vitro antiproliferative cell studies and in vivo local anti-tumor effects. In total, the paper is well organized and the discussion is convincing. And there are several suggestions/questions are below:
1、To be a beautiful review, pay attention to some details in the paper format, such as the size and font in Figures, table standardization, etc. Such as Figure 1, Figure 2, Figure 3, and Table 1.
2、The review picture and table are too single, and lack contrast and richness, please enrich the polymer in the electrospinning film, the drug loaded on the electrospinning film, therapeutic effects, and other related pictures.
3、Four typical polymers are mentioned, and it is suggested that a table be provided to discuss the similarities and differences between the four polymers themselves and with other polymers.
4、How is the clinical translation of the chemotherapy drug-loaded electrospun membranes? How many products are under clinical translation?
5、Some related research about drug delivery polymers should be cited to highlight the potential applications of these materials. Biomolecules 2022, 12, 636.; Biomater. Sci., 2022,10, 5369-5390
Author Response
Response to the reviewer's comments
Response to Reviewer #1
1. To be a beautiful review, pay attention to some details in the paper format, such as the size and font in Figures, table standardization, etc. Such as Figure 1, Figure 2, Figure 3, and Table 1.
Response: Thanks for the advice. All the fonts and size in the figures and tables have been unified with the Palatino Linotype font of MDPI journals.
2. The review picture and table are too single, and lack contrast and richness, please enrich the polymer in the electrospinning film, the drug loaded on the electrospinning film, therapeutic effects, and other related pictures.
Response: Thanks for the advice. An extra figure depicting the abovementioned aspects was put into the article as figure 3 (Page 7, Line 284).
3. Four typical polymers are mentioned, and it is suggested that a table be provided to discuss the similarities and differences between the four polymers themselves and with other polymers.
Response: Thanks for the advice. An extra table comparing varied polymers discussed was put into the article as table 1 (Page 5, Line 205).
4. How is the clinical translation of the chemotherapy drug-loaded electrospun membranes? How many products are under clinical translation?
Response: Thanks for the reviewer’s professional suggestions. We searched the relevant content carefully. Unfortunately, there are not yet any FDA-approved products of electrospun chemo-drug-loaded membranes on the U.S. market. There were currently two electrospun material products listed on the clinical trial database of the U.S. national institute of health. One is an antibacterial electrospun material loaded with ciprofloxacin (CIP), metronidazole (MET), and minocycline (MINO) for immature necrotic teeth. The other is a tissue repair scaffold for treating dermatologic wounds created by the surgical removal of non-melanoma skin cancers. Hopefully, there will be chemo-drug-loaded electrospun material products for clinical trials against local carcinoma. This extra explanation was arranged in the therapeutic effects section to discuss clinical trial stage chemo-drug loaded electrospun membranes.
5. Some related research about drug delivery polymers should be cited to highlight the potential applications of these materials. Biomolecules 2022, 12, 636.; Biomater. Sci., 2022,10, 5369-5390
Response: Yes, we found these articles helpful to increase the robustness of the introduction part and cited them in references #3, and #4 (Page 1, Line 45).
Besides these routine procedures, drug-loaded hydrogels, electrospun membranes, and other in situ delivery systems also have great potential against local carcinoma since they are excellent in local tissue adhesion with high porosity that facilitate drug delivery and cell attachments[3,4].
- Tang, Y.; Varyambath, A.; Ding, Y.; Chen, B.; Huang, X.; Zhang, Y.; Yu, D.-g.; Kim, I.; Song, W. Porous organic polymers for drug delivery: hierarchical pore structures, variable morphologies, and biological properties. Biomater Sci-Uk 2022, 10, 5369-5390, doi:10.1039/D2BM00719C.
- Zhang, Y.; Song, W.; Lu, Y.; Xu, Y.; Wang, C.; Yu, D.-G.; Kim, I. Recent Advances in Poly(α-L-glutamic acid)-Based Nanomaterials for Drug Delivery. Biomolecules 2022, 12, doi:10.3390/biom12050636.

Reviewer 2 Report
Dear authors
This is an interesting review related to the release profiles and efficacies of chemotherapy drug-loaded electrospun membranes. However, it must be revised significantly based on the following major problems
1. The abstract does not present the clear objectives of this review. Moreover, the main contents of the review also need to describe briefly and precisely for attracting attention of readers
2. Figure 1, the last images are not labelled. Please add it and describe in the caption
3. Section 2.1.1. and 2.1.2, the authors simply describe polymers and techniques used to preparation of membranes. It is better to summarize and compare them in the tables. I think that each section should summrize in one table.
4. Problems in development of chemotherapy drug-loaded electrospun membranes and potential strategies for improving their release efficacies need to point out and discuss
5. Please check all abbriviations in this manuscript. Some of them are missed a full expression such as GLOBOCAN, PLGA, ....
6. Figure 2 caption shoule be described more detail a, b, c, d
7. Schemes and graphs in Figure 3 should be labelled and discribed in detail
Author Response
Response to the reviewer's comments
Response to Reviewer #2
1. The abstract does not present the clear objectives of this review. Moreover, the main contents of the review also need to describe briefly and precisely for attracting attention of readers.
Response: Thanks for the advice. The abstract was fixed with new sentences focusing on the aim of this review. The main content was optimized to be more straightforward and more precise.
Abstract: Electrospun fibrous membranes loaded with chemotherapy drugs have been broadly studied, many of which had promising data demonstrating therapeutic effects on cancer cell inhi-bition, tumor size reduction, life extension of tumor-bearing animals, and more. Nevertheless, their drug release profiles are difficult to predict since their degradation pattern varies with crystalline polymers. In addition, there is room for improving their release performances, optimizing the release patterns, and achieving better therapeutic outcomes. In this review, the key factors affecting elec-trospun membrane drug release profiles have been systematically reviewed. Case studies of the release profiles of typical chemotherapy drugs are carried out to conclude the preferred polymer choices and techniques to achieve expected prolonged or enhanced release profiles. The therapeutic effects of these electrospun chemo-drug-loaded membranes are also discussed. This review aims to assist the design of future drug-loaded electrospun materials to achieve preferred release profiles with enhanced therapeutic efficacies.
2. Figure 1, the last images are not labelled. Please add it and describe in the caption
Response: Thanks for pointing out this problem in manuscript. The labels are added with captions updated (Figure 1, Page 2, Line 95).
3. Section 2.1.1. and 2.1.2, the authors simply describe polymers and techniques used to preparation of membranes. It is better to summarize and compare them in the tables. I think that each section should summrize in one table.
Response: Thanks for the advice. An extra table comparing and summarizing various polymers discussed in the section was put into the article as table 1 (Page 5, Line 205).
4. Problems in development of chemotherapy drug-loaded electrospun membranes and potential strategies for improving their release efficacies need to point out and discuss
Response: Thanks for the advice. This discussion is important for the review, and sorry for the miss. This part was added as an extra section 2.2.1 in the article to review the problems/challenges of developing membranes with proper release properties. Then sections 2.2.2 and 2.2.3 explained the strategies for improving release profiles.
5. Please check all abbriviations in this manuscript. Some of them are missed a full expression such as GLOBOCAN, PLGA, ....
Response: All the abbreviation problems are fixed.
6. Figure 2 caption shoule be described more detail a, b, c, d
Response: Captions updated in figure 2 (Page 5, Line 209).
7. Schemes and graphs in Figure 3 should be labelled and discribed in detail
Response: The Figure 3 in old mancript has change to Figure 4 in the new version. Labels updated with more descriptions (Page 9, Line 309).
Figure.4 Key factors (polymers, drug hydrophilicity, electrospinning techniques) affecting the release profiles of chemo-drug loaded electrospun membranes. Blending or surface absorbing drugs with hydrophilicity mismatched polymers would lead to burst release with long stationary phase afterwards. Proper matching of drug-polymer hy-drophilicity or adapting proper techniques would result in gradual release with profiles close to logistic or linear increase.

Round 2
Reviewer 2 Report
It can be accepted for publication